# *LADON*, a Natural Antisense Transcript of *NODAL*, Promotes Tumour Progression and Metastasis in Melanoma

**DOI:** 10.3390/ncrna9060071

**Published:** 2023-11-15

**Authors:** Annie Dutriaux, Serena Diazzi, Chiara Bresesti, Sylvie Hardouin, Frédérique Deshayes, Jérôme Collignon, Domenico Flagiello

**Affiliations:** Institut Jacques Monod, Université Paris Cité, CNRS, F-75013 Paris, France; annie.dutriaux@ijm.fr (A.D.); serena.diazzi1@gmail.com (S.D.);

**Keywords:** lncRNA, melanoma, invasion, metastasis, exosomes, A375M, FO1, mesenchymal to amoeboid transition

## Abstract

The TGFβ family member NODAL, repeatedly required during embryonic development, has also been associated with tumour progression. Our aim was to clarify the controversy surrounding its involvement in melanoma tumour progression. We found that the deletion of the *NODAL* exon 2 in a metastatic melanoma cell line impairs its ability to form tumours and colonize distant tissues. However, we show that this phenotype does not result from the absence of NODAL, but from a defect in the expression of a natural antisense transcript of *NODAL*, here called *LADON*. We show that *LADON* expression is specifically activated in metastatic melanoma cell lines, that its transcript is packaged in exosomes secreted by melanoma cells, and that, via its differential impact on the expression of oncogenes and tumour suppressors, it promotes the mesenchymal to amoeboid transition that is critical for melanoma cell invasiveness. *LADON* is, therefore, a new player in the regulatory network governing tumour progression in melanoma and possibly in other types of cancer.

## 1. Introduction

Metastasis is responsible for up to 90% of cancer deaths [1], making the determination of the molecules involved in this process crucial to the improvement of diagnosis and treatment. The *NODAL* gene, which encodes a TGFβ family member, was identified as a possible player in the acquisition of metastatic behaviour [2,3]. The NODAL ligand binds a receptor complex containing type I (ALK4 or 7) and type II (ActRIIA or B) serine/threonine kinase receptors, the activation of which leads to the phosphorylation of the signal transducers SMAD2 or SMAD3. NODAL is best known for its role during development, where it is required both to maintain the undifferentiated state of embryonic precursors and to specify the identity of specific cell types, including several motile cell types [4]. It is also expressed in the adult, notably in tissues that undergo periodic renewal or remodelling under the control of hormonal stimuli, such as the endometrium and the mammary gland [5,6]. The NODAL expression in tumour cells has been correlated with their plasticity and invasive behaviour [2,7], in line with its functions in embryonic and adult tissues.

The involvement of NODAL in tumour progression and metastasis was first described in melanoma cell lines [3], but although evidence supporting such a role for this factor has been obtained in other cancer types, its actual contribution to metastasis in melanoma has been disputed [8,9], as its expression was not always detected there. A review of available *NODAL* expression data in cancer cells suggested the existence of distinct splice variants [10], all including the second exon of the gene but sometimes, notably in melanoma cell lines, lacking its third (and last) exon and therefore lacking the capacity to produce a functional ligand. A previous study had already shown that the expression of CRIPTO, an obligatory co-receptor of NODAL, was weak in metastatic melanoma cell lines, and restricted to a small cell subpopulation [11]. These observations raised the possibility in this context of a different mode of action of *NODAL* in conferring metastatic properties to cells, independent of its known signalling function.

We therefore used genetic and pharmacological approaches, both in vivo and in vitro, to reassess the importance of *NODAL* in mediating the invasive properties of melanoma cell lines. Our results argue that the NODAL protein plays no role, but that a natural antisense transcript, which we have named *LADON*, and which overlaps with the *NODAL* exon 2, promotes tumour progression and invasion. Our study identifies *LADON* as a novel regulator of melanoma progression.

## 2. Results

### 2.1. NODAL Exon 2 Promotes Tumour Growth and Metastasis in A375 Cells

We used RT-PCR analysis to characterize the *NODAL* expression in a panel of human cell lines relevant to cancer and melanoma: melanocytes, non-metastatic melanoma (MNT1), metastatic melanoma (A375M–noted A375 in the rest of the study, FO1, 888 Mel, SLM8), breast cancer (MCF7), and embryonic kidney (HEK293). A primer pair specific for the *NODAL* exon 2 amplified the expected band in all of them (Figure 1A and Appendix A), whereas primers spanning the junction of exons 2 and 3 detected a transcript only in the embryonic kidney cell line (Figure 1A and Appendix A). The primers spanning the junction of exons 1 and 2 also failed to detect a band in all cell lines, except in the HEK293 line (Appendix A). These results are consistent with studies that have detected the expression of the *NODAL* exon 2, but not the *NODAL* exon 3, in melanoma cell lines [8,10], and further suggest that the *NODAL* exon 1 is not expressed either.

To test for the possible contribution of the *NODAL* exon 2 to the properties of melanoma cells, we used genome editing to delete it in all copies of the gene present in A375 cells (Figure 1A). This exon encodes more than half of the mature signalling protein and its deletion prevents the production of a functional ligand [12]. Five of the independent mutant clones thus obtained, designated A375ΔE2a to e, were characterized (Appendix A). The absence of the *NODAL* exon 2 was found to impair their ability to close the gap in a 2D wound-healing (scratch) assay (Figure 1B and Appendix A). This effect was detected in less than 12 h of culture, which suggests that it results from a decrease in cell motility, not a decrease in mitosis. Measurements allowed us to plot the respective gap closure rates of mutant and unmodified A375 cells (Figure 1C). Exposure to the exogenously supplied recombinant NODAL or ACTIVIN, another TGFβ-related ligand that signals via the same receptors as NODAL, did not rescue the motility of the A375ΔE2 cells in the same assay, and did not affect that of unmodified A375 cells either (Figure 1C and Appendix A). Treatment with the SB431542, a pharmacological inhibitor of the ACTIVIN/NODAL type I receptors ALK4, 5 and 7, also had no effect (Figure 1C), confirming that ACTIVIN/NODAL signalling is not involved in whatever is determining gap closure rates.

To assess the impact of the mutation on tumour progression and metastasis in vivo, chorioallantoic membrane (CAM) assays [13] were performed in fertilized chicken eggs using unmodified A375 cells and A375ΔE2d mutant cells (Appendix A). 1.10^6^ cells were deposited on the upper CAM of E9 chick embryos. Tumour progression was assessed 9 days later (E18) by two criteria: (1) by measuring the weight of the tumour formed on the upper CAM; and (2) by measuring the presence of human cells in the lower CAM and the chick embryo liver. The A375ΔE2d mutant cells gave rise to tumours that were only about 1/9 the size of those derived from unmodified A375 cells (Figure 1D,E), and their metastatic presence in the lower CAM and the liver was reduced to 1/10 and 1/2 of the levels seen with the unmodified cells, respectively (Figure 1F). Histological analyses showed that primary tumours derived from mutant A375ΔE2 cells had a higher mitotic index but were also far more necrotic than those derived from unmodified cells (Figure 1D and Appendix A).

These results indicate that A375 cells require the *NODAL* exon 2 to promote normal tumour growth and invasion. However, the lack of response of the cells to either the exogenous NODAL and ACTIVIN or to an inhibitor of their signalling pathway, as well as the absence of full-length *NODAL* transcripts, strongly argue that the mode of action of the gene in this context does not rely on the production of the mature ligand.

### 2.2. A Natural Antisense Transcript of NODAL Exon 2 Is Expressed in Melanoma Cells

An update of the *Ensembl* database (GRCh38.p13 primary assembly) revealed the presence of a third transcript at the human *NODAL* locus, in addition to the two *NODAL* splice variants already known. This 1728nt long RNA (AC022532; ENSG00000280401) is transcribed from the plus strand (opposite to *NODAL*) and is annotated as non-coding (Figure 2A and Appendix A). The corresponding transcription unit is hereafter designated as *LADON*. A full-length cDNA was first obtained from a human teratocarcinoma cDNA library [14]. It starts in *NODAL* intron 2 and ends in *NODAL* intron 1 and therefore includes a sequence complementary to that of the entire *NODAL* exon 2. Its 5′ and 3′ regions contain short interspersed nuclear elements (SINEs) Alu and MIR. Homologous transcripts of similar size were found in a chimpanzee and an orangutan (CK820-G0007942 and CR201-G0049649, respectively). A longer homolog has been found in pigs (ENSSSCG00000051352), but none has been reported in mice and rats, nor in any of the other classical vertebrate animal models (zebrafish, xenopus, and chick), which may suggest that it arose within the mammalian lineage.

To find out whether changes in the *LADON* expression correlate with tumour progression and prognosis in humans we used the GEPIA web server to analyze transcriptomic data collected by The Cancer Genome Atlas (TCGA) and Genotype-Tissue Expression (GTEx) projects [15]. Dramatic differences in the *LADON* expression levels between tumours and their corresponding healthy tissues were detected in a majority (21/31) of the cancer types analyzed, with the largest differential—a 96% reduction in normal expression level—found in skin cutaneous melanoma (SKCM; Figure 2B). However, if we look at how this expression varies with the stage of the cancer, the data show that it peaks at stage II when the tumour is growing locally and preparing to metastasize (Figure 2C). Finally, SKCM patients with higher levels of *LADON* expression survived longer than patients with lower expression levels (panel Figure 2D). No copy number alterations were found at the *NODAL* locus in these patients.

These data are consistent with the possibility that *LADON* plays a role in melanoma progression and metastasis, but the observation that higher expression levels delay cancer progression was at first difficult to reconcile with our in vivo results.

### 2.3. Metastatic Melanoma Cell Lines Share the Ability to Upregulate LADON

To confirm the presence of *LADON* transcripts in A375 cells, a total of A375 RNA was reverse transcribed using either a reverse (N2R) or a forward (N2F) primer concerning the orientation of its transcription (Figure 1A). The resulting cDNAs were used to PCR-amplify sequences spanning the exon 2 and the adjoining 5′ or 3′ regions (Figure 2A). The fact that a band of the expected size was only obtained with cDNAs transcribed using the reverse primer N2R confirmed the expression of *LADON* and the absence of *NODAL* transcripts (Appendix A).

We then used a specific pair of primers (L1F-L1R, Figure 2A) to detect *LADON* transcripts, and amplified a band of the expected size in all the cell lines of our panel (Figure 2E). The RT-qPCR analysis of *LADON* expression in melanocyte and melanoma cell lines revealed big variations in the expression of *LADON* within individual metastatic melanoma cell lines but not in the melanocyte and non-metastatic cell lines. The extent of these variations was correlated with the length of time the cells had been cultured, as a 2- to 4-fold increase in the *LADON* expression was observed between 24 h and 96 h after seeding the culture (Figure 2F). However, the basal level of *LADON* expression was not predictive of the metastatic or non-metastatic nature of the cell lines in our panel (Figure 2G). This basal level was 5- to 20-fold higher in keratinocytes than in melanoma cell lines, which at least partly explains the high level of the *LADON* expression detected in skin tissue (Figure 2B). Interestingly, A375ΔE2 clones, which expressed a truncated *LADON* transcript (*LADON-ΔE2*) reduced to the 5′ and 3′ parts that flank exon 2 (Figure 2A and Appendix A), showed no sign of increased expression after 96 h in culture (Appendix A), implying that the deletion deprived *LADON* of a critical regulatory input.

These results confirm that the absence of *NODAL* did not cause the changes in behaviour we have characterized in cells deleted for the *NODAL* exon 2. They identify *LADON* as a candidate and suggest a correlation between the propensity of metastatic cell lines to upregulate its expression and their ability to adopt more aggressive behaviour. Such a correlation would be in line with the expression data obtained in SKCM patients.

### 2.4. Melanoma Cells Secrete Factors That Promote LADON Expression

To investigate what is triggering the increase in *LADON* expression in the A375 cell line, we first tested its dependence on cell density [16]. The *LADON* transcript levels measured after 24 h culture in cells seeded at high (80%) density were actually no different from those measured in cells seeded at low (20%) density (Figure 3A). The increase in *LADON* expression found after 72 h culture was therefore more likely to depend on the duration of the culture. To determine whether the *LADON* expression is dependent on secreted factors accumulating in the culture medium, A375 cells were grown in standard medium (SM), with or without 10% fetal calf serum (FCS), and the media were collected 3 days later when the cells had reached high density. These A375-conditioned media (CM) were then used to culture cells seeded at low density for 24 h. For A375 cells, this resulted in a 2.4-fold increase in the expression of *LADON* (Figure 3B), compared to its expression level when cultured in SM. This increase occurred regardless of the presence of FCS, thus eliminating nutrient depletion as a possible cause (Appendix A). Other metastatic melanoma cell lines, FO1 and 888 MEL were also found to increase the *LADON* expression when cultured in their own CM (Appendix A). In contrast, cells from non-tumorigenic non-metastatic cell lines exposed to their own CM did not affect the *LADON* expression (MNT1) or even reduce it (DAJU, SK28). Interestingly, the MNT1 cells did not respond to A375-CM, but the MNT1-CM did trigger an increase in *LADON* expression in A375 cells (Figure 3B). Thus, while both the A375 and MNT1 cells secrete and accumulate *LADON*-inducing factors in their culture medium, only the A375 cells are able to respond to these factors, suggesting that this ability is specific to metastatic cell lines.

Analysis of the published content of the A375 cell-derived exosomes (GSE35388) [17] revealed a relative enrichment in the *LADON* transcripts (Appendix A). This raised the possibility that secreted exosomes contribute to the inductive capacity of the CM. We purified exosomes from the A375-CM and confirmed that, compared with cells, they show a 7-fold enrichment in the quantity of the *LADON* transcripts (Appendix A). However, neither these purified exosomes nor the supernatant left after their separation could on their own induce an increase of the *LADON* expression in A375 cells similar to that obtained with A375-CM (Figure 3C). It is only after these two separate fractions were combined that such a response was obtained, suggesting that this induction requires factors from both fractions. We then found that the exon 2-depleted A375ΔE2-CM had the same capacity to induce *LADON* expression as that obtained from the A375 cells (Figure 3D), implying that a full-length *LADON* is not required in the CM to increase its own expression and that the production of *LADON*-inducing factors is *LADON*-independent. Cells from the A375ΔE2 clones, however, failed to increase the expression of the *LADON*-Δ*E2* transcript when exposed to the same CM (Figure 3D), confirming that the capacity of the locus to respond is dependent on its integrity.

### 2.5. LADON Expression Promotes Invasion

Migrating melanoma cells rely on two inter-convertible modes of migration, designated mesenchymal and amoeboid [18,19] with the faster amoeboid mode being the most efficient to drive invasion [20,21,22]. We therefore investigated the effect of the A375-CM on cell morphology and invasiveness. The mesenchymal to amoeboid transition (MAT), which involves a change from an elongated cell to a smaller and rounded cell morphology, was monitored via F-actin staining (Figure 3E) [23]. In the SM, no clear difference was seen between cultures of the mutant and unmodified A375 cell populations, both showing similar cell size ranges (Appendix A). However, when grown in the CM, the average size of the A375 cells decreased, with no cells covering an area greater than 50 μm^2^, and their roundness index increased (Figure 3F and Appendix A). When the capacity of these cells to transmigrate through a collagen layer was quantified over 24 hours, this change in the composition of the population was associated with a 7-fold increase in the rate of transmigration (Figure 3G and Appendix A). The FO1 cells grown in their own CM showed a similar decrease in mean cell size and an increase in their transmigration rate (Appendix A and Figure 3H). In contrast, the A375ΔE2 cells, whose transmigration rate in SM was one-tenth that of the A375 cells, showed no improvement in transmigration capacity when cultured in the CM (Figure 3G and Appendix A). *LADON* is therefore endowing the A375 cells with the capacity to undergo CM-induced changes in cell behaviour. Again, to assess the contribution of the exon 2 sequences to this particular effect of the CM we analyzed the effect of the A375ΔE2-CM on the A375 cells. This treatment had no significant impact on the cell size of the A375 cells (Figure 3F) and although their transmigration rate showed a slight increase, it was about a quarter of that of cells treated with the A375-CM (Figure 3G and Appendix A). This suggested that, unlike the increase in *LADON* expression, changes in cell shape and behaviour are induced by an exon 2-dependent component of the CM. The A375 cells treated with the A375ΔE2-CM would therefore have to increase their expression of *LADON* before they could secrete this component themselves and respond to it. It follows that after just 24 h (Figure 3C,F,G) their response could be delayed, but that given sufficient time they should undergo the same changes as those treated with the A375-CM, and that the differences observed between them should disappear. This is indeed what we observed when we repeated the experiment but cultured the cells for 48 h instead of 24 h (Appendix A).

To confirm that it is the mutation of the *LADON* transcript and not another unrelated consequence of the deletion that is causing the drop in invasiveness, we examined the behaviour of the A375 and FO1 cells following siRNA-mediated depletion (knockdown) of the *LADON* expression. Three different siRNAs located at the 5′ end of the gene were tested. The RT-PCR results showed a significant decrease in *LADON* expression with two of them, used alone or together (Appendix A). The knockdown (KD) of *LADON* severely impaired the ability of the A375 and FO1 cells to transmigrate through a collagen monolayer over 24 h, behaviour reminiscent of the A375ΔE2 clones (Figure 3I). We also noticed that in both cell lines *LADON* KD cells systematically reached confluence earlier than controls. They indeed had higher cell counts at the end of the culture (Appendix A), suggesting that their proliferation rate was positively affected. We confirmed that the A375ΔE2 clones have higher proliferation rates than their parental cell line (Appendix A), an observation consistent with the histological analysis of the primary tumours they gave rise to in the CAM assay (Appendix A). Finally, to confirm that *LADON* promotes invasion independently of *NODAL* we assessed whether its forced expression in the A375ΔE2 cells could rescue their transmigration rate. Transient transfection experiments of cells from two mutant clones indeed showed a marked improvement in the transmigration rates of those that were transfected with a plasmid expressing *LADON* (Figure 3J).

Taken together, these results demonstrate that the expression of *LADON* in melanoma cells conditions both their capacity to produce factors that promote changes conducive to metastasis (i.e., MAT, increase in invasiveness, decrease in proliferation), and their capacity to respond to such factors.

### 2.6. LADON Affects the Expression of Oncogenes and Suppressors of Tumour Progression

To identify factors acting downstream of *LADON*, we used mass spectrometry to compare the proteomes of the A375 and A375ΔE2 cells (volcano plot, Figure 4A). We focused on proteins that were identified by at least 2 independent peptides and with a fold change superior to 1.5. A Gene Ontology (GO) term analysis of the 28 proteins thus obtained showed significant enrichment in components of filamentous actin (PDLIM4, FERMT2, and NCKAP1), stress fibers (PDLIM4, Dynactin, and FERMT2), as well as lamellipodia (PDLIM4, PLCG1, FERMT2, and NCKAP1) (Figure 4A; Table 1). All of these were upregulated in the A375ΔE2 cells, suggesting a repressive role of *LADON* in their expression. Among the six proteins that had a fold change superior to 2 (Table 2), 5 were overexpressed in the A375ΔE2 cells and turned out to be known tumour (FAM107B, ACPP) or metastasis (NDRG1) suppressors [24,25,26,27], or transcription factors known to activate these proteins (p65, also known as NF-κB regulatory subunit RELA, is an activator of ACPP) [28]. Conversely, the under-expressed protein PPP4R2 is a regulatory subunit of the protein phosphatase 4 complex (PPP4), which has been linked to tumour progression in several cancers [29,30].

Together, these results suggest that *LADON* promotes the MAT and cell motility in the metastatic melanoma cell line A375 at least partially via its impact on the expression of known promoters and suppressors of tumour progression.

## 3. Discussion

Our investigations found no evidence that NODAL could influence the behaviour of metastatic melanoma cells but revealed instead their reliance on the expression of *LADON*, a natural antisense transcript of the *NODAL* locus. Our findings support a critical role for its transcript in the network of interactions that governs tumour progression and metastasis in melanoma, notably in the mechanisms underlying how cells produce and respond to pro-metastatic signals.

The absence of the *NODAL* transcripts in melanoma cell lines reported here is consistent with the results of other studies [8,10], as well as with a recent characterization of transcript diversity at the human *NODAL* locus [31]. This latter report and our own study show that most exon 2-containing RNA species present in melanoma cells are in fact transcribed from the strand opposite to *NODAL* and correspond to *LADON*. This finding explains some of the inconsistencies found in previous studies of the role of *NODAL* in melanoma, as an exon 2-based assay was used to track the expression of the gene in several of them [10,11,31,32]. The error of confusing *LADON* expression with *NODAL* expression was compounded by the use of commercial antibodies cross-reacting with non-specific proteins that were mistakenly identified as NODAL, as was later demonstrated [8]. Donovan et al. showed that while there is evidence that TGFβ and ACTIVIN-A, both signalling via SMAD2,3, do promote tumour progression in melanoma in various ways, no such case can be made for NODAL.

The absence of *NODAL* expression in melanoma cells means that the changes in behaviour we characterized in the A375 cells deleted for *NODAL* exon 2, both in vivo and in vitro, resulting from the impact of the deletion on the expression of genes other than *NODAL*. The only disturbance we could identify in the expression of the different transcription units in the vicinity of exon 2 is that of *LADON*, which expresses a truncated transcript and can no longer respond to inductive treatment. The knockdown of *LADON* in the A375 and FO1 cells results in a loss of invasiveness and an increase in cell proliferation similar to those obtained by the deletion of exon 2. Finally, the forced expression of *LADON* in exon 2-deleted cells reversed the impact of the deletion on their transmigration rate, demonstrating that the transition to a more invasive behaviour is dependent on the *LADON* transcript itself, acting independently of *NODAL*. These results, in line with its impact on the proteome of melanoma cells, show that *LADON* is involved in the transition from a proliferative cell identity to a less proliferative but more invasive cell identity, a change of phenotype inherent in the metastatic process in melanoma [33,34,35].

An interesting aspect of *LADON*’s involvement in melanoma is that it seems to require tight regulation of its expression. Our own results show that melanoma cells are at first expressing much lower levels of the transcript than healthy skin cells, but that this expression reaches a peak at stage II before going back to low levels. This observation finds some echo in the fact that the capacity to increase *LADON* expression correlates with a metastatic identity in our panel of melanoma cell lines.

There are now multiple examples of lncRNAs being involved in tissue physiology and in the regulation and misregulation of cellular processes, which often result in cancer [36,37,38]. This is a major reason for the interest in lncRNAs as biomarkers for diagnosis and prognosis, but also as potential targets for therapy or as tools to correct defective gene expression underlying cancer and other pathologies [39,40]. Even at its peak, the expression of *LADON* never reaches a level that would allow it to be an effective miRNA sponge [41,42]. The low level of expression of *LADON* is actually a common feature of lncRNAs and recent evidence suggests that it may be a key determinant of their specificity [43,44]. An interesting feature of the transcript is the presence in its 3′region of the short interspersed elements (SINEs) Alu and MIR. The presence of Alu elements in lncRNAs has been associated with the capacity of such transcripts to regulate RNA transcription, decay, and splicing [45,46,47], while the presence of other SINEs has been associated with the regulation of other processes such as translation [48,49]. In that respect, it will be interesting to investigate whether or not the changes in protein expression highlighted by our mass spectrometry analysis are preceded by changes in the expression of the corresponding genes.

*LADON* expression has so far only been reported in melanoma and breast cancer cell lines [31], and this study, but it is likely to be present in other cancer cell lines. There is evidence that it is present in healthy human cell types and tissues, where we currently have no indication of its function. Most of these cells, whether cancerous or not, do not express *NODAL*. For those that do express *NODAL*, future studies should investigate whether *LADON* has the ability to modulate NODAL signalling.

## 4. Materials and Methods

### 4.1. Plasmids

pcDNA3.1 and GFP pcDNA 3.1 are from Thermo Fisher Scientific (Waltham, MA, USA). LADON corresponding to AK001176.1, 1725bp was synthesized and inserted in pCDN3.1 (Genscript, Leiden, The Netherlands). This vector was used for the transfection of the A375ΔE2 a and d for rescue experiments. 

### 4.2. Cell Lines and Cell Culture

The melanoma cell line A375M was purchased from the ATCC. The normal melanocyte cell line was kindly provided by Nathalie Andrieu (Centre de Recherches en Cancérologie de Toulouse). The SLM8 cell line, kindly provided by M Viguier, is derived from lymph node metastasis. The MNT1 and FO1 melanoma cell lines were a gift from Lionel Larue (Institute Curie CNRS UMR3347, INSERM U1021, Institute Curie). The 888 Mel cell line, a gift from A. Mauviel (Institut Curie/CNRS UMR 3347/INSERM U1021), is derived from the lung metastatic WM793 melanoma cell line. Cells were grown in DMEM/F12 Glutamax (Table 3, Invitrogen, Cergy-Pontoise, France) supplemented with antibiotics and 10% fetal calf serum (FCS) in a 5% CO_2_ atmosphere. 

To prepare the conditioned medium, melanoma cell lines were seeded at 70% confluence and then incubated for 72 h in DMEM medium with or without FCS, as stated. This melanoma-conditioned medium was collected, centrifuged at 11,000× *g* for 15 min to remove cell debris, and stored at −80 °C.

The source and identifier of all the plasmids and cell lines are listed in the reagent and resource Table 3.

### 4.3. Transmigration Assay

Melanoma cells (2 × 10^5^) were seeded on the upper compartments of 2 mg/mL type I collagen-coated culture inserts (8 µm pores -Greiner Bio-One SAS, Courtaboeuf, France). The DMEM supplemented with 10% FCS was used as a chemoattractant. The A375 and A375ΔE2 cells were allowed to migrate at 37 °C and 5% CO_2_ for 24 h. The non-migrating cells on the upper face of the filter were removed by gently scraping them off using a cotton swab. The cells on the lower face were washed in PBS, fixed with 4% formaldehyde for 10 min, and washed in PBS. The nuclei were then labeled with Hoechst for 5 min and washed again. The migrating cells were observed under an epifluorescence microscope using a 10× magnification. Five to fifteen pictures of adjacent fields of the central zone of each Transwell were taken. Fluorescence was quantified with the IMAGEJ software [50] (US National Institutes of Health, Bethesda, MD, USA). Histograms display the data obtained from three independent experiments, and all the experiments were performed in duplicate. *p*-values were calculated using an ANOVA. Data are displayed as normalized results, where the nuclei of melanoma cells are set to 1 (or 100%) when grown in control conditions.

### 4.4. A375 Staining

For immunofluorescence experiments, the A375 cells were seeded onto glass coverslips in twelve-well plates and incubated overnight. Next, the cells were fixed in a 4% formaldehyde solution and incubated with Alexa Fluor 647-coupled phalloidin to visualize F-actin, and Hoechst 33342 to label nuclei. Coverslips were then further washed in PBS, treated with Fluoromount-G anti-fade (Southern Biotech), and analyzed by confocal microscopy. The shape of the A375 cells (n > 20), elongated or round was analyzed by the IMAGEJ software, and the proportion of each corresponding cell type was quantified.

### 4.5. Wound Healing/Scratch Assay

Cell migration was examined using a wound-healing assay. In brief, 0.2 × 10^6^ cells were seeded in a well of twelve-well plates, and at the confluence, a scratch wound was made with a 10 µL pipette tip and then washed twice with PBS to remove cell debris. Wells were photographed under phase-contrast microscopy (time = 0) while cells were allowed to migrate into the scratch wound area for up to 18 h at 37 °C 5% CO_2_ atmosphere using an Essenbio IncuCyte apparatus. The speed of the closure area was calculated over time by the IMAGEJ software. The data are represented as a ratio of migratory cells normalized to A375 cells migration in the control condition. All the experiments were performed in duplicate.

### 4.6. CRISPR/CAS9 Genome Editing

The CRISPR/Cas9 genome editing was performed with the GeneArt CRISPR Nuclease Vector Kit according to the manufacturer’s instructions (Life Technologies). To select the target sequence for genome editing, the genomic sequences surrounding the *NODAL* exon 2 were submitted to an online CRISPR Design Tool (https://portals.broadinstitute.org/gppx/crispick/public (accessed on 9 October 2023)) Two target sites were selected upstream and downstream of this sequence. The oligonucleotides used to construct the gRNAs for the human *NODAL* gene exon 2 (deletion of 698 pb) are listed in Appendix A.

The ds oligonucleotides generated were cloned into the GeneArt CRISPR Nuclease Vector. Competent E. coli cells were transfected with 3 μL of the ligation reaction, and then 50 µL from the transformation reaction was spread on a pre-warmed LB agar plate containing 100 µg/mL ampicillin. The plates were incubated overnight at 37 °C. The identity of the ds oligonucleotide insert in positive transformants was confirmed by sequencing. For the A375 cell line transfection, the cationic lipid-based Lipofectamine 2000 Reagent was used. The A375 cells positive for the transfection were sorted by FACS using OFP, a fluorescent protein present in the GeneArt CRISPR Nuclease Vector. Genotyping was performed on 96 clones to detect the deletion. Primers used to assess the efficacy of the CRISPR/Cas9 deletion are listed in Appendix A.

Length of non-deleted amplicon: 1141 pb.

Length of deleted amplicon: 445 pb.

### 4.7. siRNAs

siRNAs against LADON (two independents) and negative-control RNA were chemically synthesized (Dharmacon Research, Lafayette, CO, USA). Synthetic siRNAs were transfected with Ribocellin Transfection Reagent (Eurobio) according to the manufacturer’s instructions.

### 4.8. RNA Extraction, Reverse Transcription (RT) and Quantitative PCR (Q-PCR)

For PCR analysis, 10^5^ transfected GFP-positive cells were sorted by FACS analysis and collected into RNAse-free tubes. RNA from different cell lines was extracted with a Macherey Nagel kit for RNA purification. The first strand cDNA was synthesized with the SuperScript VILO cDNA synthesis kit (Thermo Fisher Scientific). RT-PCR amplification mixture (10 μL) contained 1/20 of cDNA product, 10X SYBR Green I Master Mix buffer, and 10 nM forward and reverse primers. Reactions were run on a Light Cycler PCR apparatus (Roche, Basel, Switzerland). Cycle conditions were 10 min at 95 °C and 45 cycles at 95 °C for 10 s, 61 °C for 10 s and 72 °C for 10 s. Each assay included a standard curve of three serial dilution points of A375 cDNA (ranging from 100 ng to 1 ng). All PCR efficiencies were above 95%. For each gene and a given RT-PCR, values were normalized to the level of expression of the reference genes *RLP13* and *GAPDH*. No significant differences in the final ratios were found between the two reference genes; therefore, only *RPL13* was used for normalization. Primers and annealing temperatures for all genes are indicated in Appendix A. For each gene, the values were averaged over at least three independent measurements.

Sequences for all the siRNAs and oligos used in this study can be found in Appendix A.

### 4.9. CAM Assay

All the elements of this assay were carried out at INOVOTION (Société: 811310127), La Tronche-France. According to decree n°2013-118 of French legislation, 2010/63/UE European directive and Animal Welfare Act of United States, all assays used or provided by INOVOTION are not considered animal testing. Fertilized White Leghorn eggs were incubated at 37.5 °C with 50% relative humidity for 9 days. More than 15 eggs were processed for each experimental condition. At stage E9, the chorioallantoic membrane (CAM) was lowered by drilling a small hole through the eggshell into the air sac, and a 1 cm^2^ window was cut in the eggshell above the CAM. A375 and A375ΔE2d cells were cultivated in DMEM/F12 glutamax medium supplemented with 10% FBS and 1% penicillin/streptomycin. On day E9, cells were detached with trypsin, washed with a complete medium, and suspended in graft medium. An inoculum of 1.10^6^ cells was added onto the CAM of each egg (E9) and then eggs were randomized into groups. On day E18, to quantitatively evaluate tumour growth, the upper portion of the CAM was removed, washed with PBS buffer, and fixed in PFA for 48 h. After that, tumours were carefully cut away from normal CAM tissue and weighed. On day E18 to evaluate the number of metastatic cells, a 1 cm^2^ portion of the lower CAM and liver were collected, 8 samples per group (*n* = 8). Genomic DNA was extracted from the CAM and liver samples (commercial kit) and analyzed by qPCR with specific primers for human Alu sequences. Calculation of Cq for each sample, mean Cq, and relative amounts of metastases for each group were directly managed by the Bio-Rad^®^ CFX Maestro software. Non-injected eggs were also evaluated in parallel, as a negative control for specificity. A one-way ANOVA analysis with post-tests was performed on the data.

### 4.10. Histology

Also carried out at INOVOTION, a total of 10 tumours (5 per genotype) were harvested, fixed in 4% formaldehyde for 48 h, trimmed, and embedded in paraffin. Paraffin blocks were sectioned at approximately 4 microns thickness. The sections were put on glass slides and stained with Hematoxylin & Eosin (H&E). Stained tumour sections were examined by an expert pathologist. Ten non-overlapping fields (HPF) from each tumour section were assessed and graded using a semi-quantitative scoring system for several parameters. Pictures were taken using an Olympus microscope (BX60, serial NO. 7D04032) and its camera (Olympus DP73, serial NO. OH05504).

The Mitotic index of each tumour was determined by counting the number of mitoses in 10 non-overlapping non-necrotic fields (objective magnification X40): 0: No mitotic figures. 1:1 mitotic figure. 2:2–3 mitotic figures. 3:>3 mitotic figures. Tumor Necrosis scores were obtained at objective magnification X10: 0: No necrosis. 1: Small foci of necrosis or scattered 2: Necrosis in less than 50% of the tumour area 3: Necrosis in 50–75% of the tumour area. 4: Necrosis in 75–90% of the tumour area. 5: Necrosis in >90% of the tumour area.

### 4.11. LC-MS/MS Acquisition

For proteomics analysis, two A375ΔE2 clones (a and d) and two parental clones (A375a and b) were used.

Protein extracts (60 µg) were precipitated with acetone at −20 °C; the protein extracts were then incubated overnight at 37 °C with 20 μL of 25 mM NH_4_HCO_3_ containing sequencing-grade trypsin (12.5 μg/mL, Promega, Madison, WI, USA). The resulting peptides were desalted using ZipTip µ-C18 Pipette Tips (Millipore, Burlington, MA, USA) and analyzed either in technical triplicates or individually by a Q-Exactive Plus coupled to a Nano-LC Proxeon 1000 equipped with an easy spray ion source (all from Thermo Scientific). Peptides were separated by chromatography with the following parameters: Acclaim PepMap100 C18 pre-column (2 cm, 75 μm i.d., 3 μm, 100Å), Pepmap-RSLC Proxeon C18 column (50 cm, 75μm i.d., 2μm, 100 Å), 300 nl/min flow rate, gradient from 95% solvent A (water, 0.1% formic acid) to 35% solvent B (100% acetonitrile, 0.1% formic acid) over a period of 98 min, followed by a column regeneration for 23 min, giving a total run time of 2 h. Peptides were analyzed in the Orbitrap cell, in full ion scan mode, at a resolution of 70,000 (at *m*/*z* 200), with a mass range of *m*/*z* 375–1500 and an AGC target of 3 × 10^6^. Fragments were obtained by high collision-induced dissociation (HCD) activation with a collisional energy of 30%, and a quadrupole isolation window of 1.4 Da. MS/MS data were acquired in the Orbitrap cell in a Top20 mode, at a resolution of 17,500 with an AGC target of 2 × 10^5^, with a dynamic exclusion of 30 s. MS/MS of the most intense precursor were first acquired. Monocharged peptides and peptides with unassigned charge states were excluded from the MS/MS acquisition. The maximum ion accumulation times were set to 50 ms for MS acquisition and 45 ms for MS/MS acquisition.

### 4.12. LC-MS/MS Data Processing

The LC-MS/MS.raw files were processed using the Mascot search engine (version 2.5.1) coupled to Proteome Discoverer 2.2 (Thermo Fisher Scientific) for peptide identification with both a custom database and the database Swissprot (from 2017) with the *Homo sapiens* taxonomy. The following post-translational modifications were searched on proteome Discoverer 2.2: Oxidation of methionine, acetylation of protein N-term, phosphorylation of serine/threonine, and phosphorylation of tyrosine. Peptide Identifications were validated using a 1% FDR (False Discovery Rate) threshold calculated with the Percolator algorithm.

### 4.13. Statistical Analysis

Data was collected and graphed using GraphPad Prism software. A 1-way ANOVA or two-sided student’s *t*-test was used to determine statistical significance where appropriate. *p*-values of less than 0.05 were considered statistically significant. Information about the statistics used for each experiment, including sample size, experimental method, and specific statistic test employed, can be found in the relevant figures or figure legends.

### 4.14. Data Collection and Analysis

The HG-U133 plus 2 arrays (Affymetrix) data presented are accessible through GEO Series accession GSE35388 analyzed by GEO2R at the National Center for Biotechnology Information Gene. 

## Figures and Tables

**Figure 1 ncrna-09-00071-f001:**
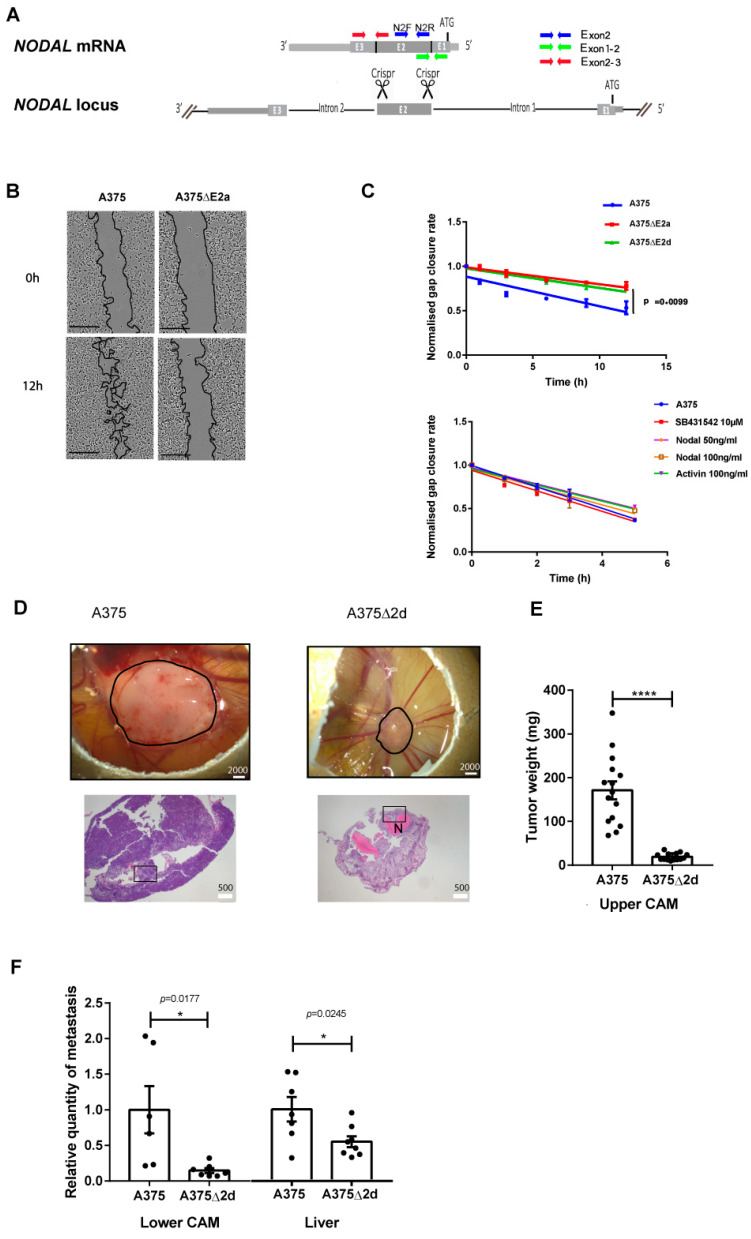
The *NODAL* exon2 promotes tumour progression and metastasis in A375 cells (**A**) Schematic of the human *NODAL* locus with its 3 exons (E1 to E3), showing the full-length *NODAL* mRNA above. The primers used to detect its expression are represented by arrows. (**B**) Representative images, acquired at t = 0 and t = 12 h, of scratch-wound healing assays performed with A375 and A375ΔE2a cells. (**C**) Top panel: Normalized gap closure rates of A375, A375ΔE2a, and A375ΔE2d cells during the scratch/wound healing assay shown in (**B**). Lower panel: Normalized gap closure rates of A375 cells treated with or without recombinant NODAL, recombinant ACTIVIN, or the ACTIVIN/NODAL signalling pathway inhibitor SB431542. The *p*-value was calculated by a two-way ANOVA test. (**D**) Top panels: Representative images of tumours (black circle) formed by A375 and A375ΔE2d cells on the upper CAM at the end of the assay (E18). The scale bar is 2000 μm. Lower panels: Representative sections of the tumours. Hematoxylin-eosin stain. Necrotic areas (N) appear in pink. Higher magnification views of selected areas (rectangles) are shown in Appendix A. The scale bar is 500 μm. (**E**) Mean weight (mg) of tumours formed by A375 (*n* = 15) and A375ΔE2d (*n* = 14) cells on top of the upper CAM 9 days after the graft. The *p*-value was calculated via a one-way ANOVA test, **** *p* < 0.0001. (**F**) Relative presence of A375 and A375ΔE2d cells in the lower CAM (n = 7 for both cell types) and in the liver (*n* = 7 and *n* = 8, respectively), as measured by the qPCR detection of human Alu sequences. The *p*-value was calculated via a one-way ANOVA test * *p* < 0.05.

**Figure 2 ncrna-09-00071-f002:**
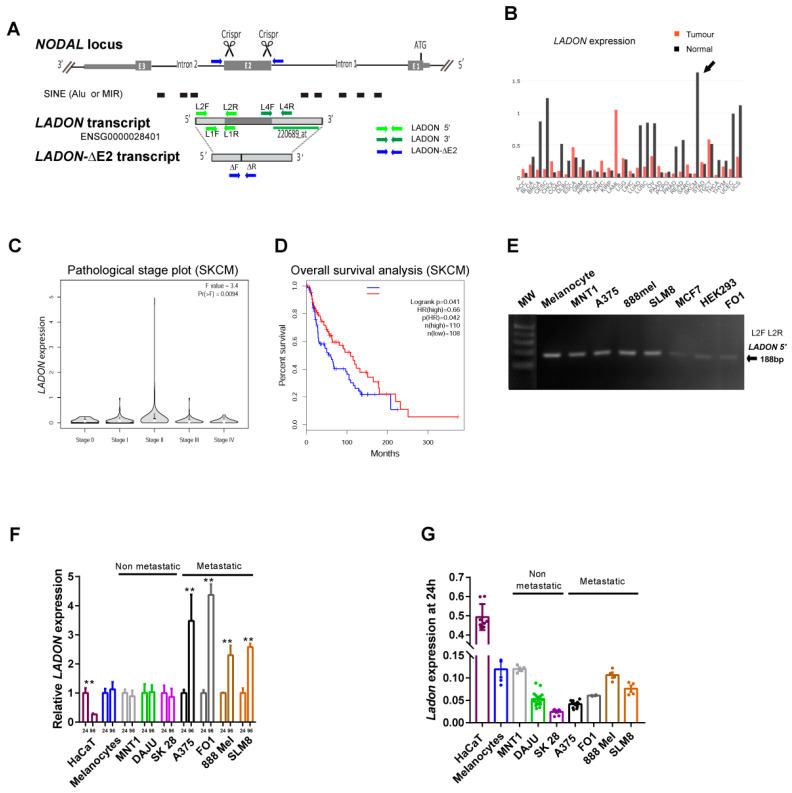
*LADON*, a natural antisense transcript of *NODAL* is a candidate oncogene in human melanoma. (**A**) Schematic of the human *NODAL* locus with its 3 exons (E1 to E3), showing the *LADON* transcript transcribed from the opposite strand below. The truncated *LADON-ΔE2* transcript is expressed in A375 cells deleted for the *NODAL* exon2. The arrows represent the primers used to track these transcripts. The black boxes show the location of short interspersed nuclear elements (SINE), which are mostly Alu and MIR sequences. (**B**) The bar plot of the *LADON* expression profile across all tumour samples and paired normal tissue was obtained from TCGA and GTEx data. The height of the bars represents the median expression of tumour types and normal tissues. The tumour abbreviation chart is provided in Appendix A. The arrow indicates the high differential of *LADON* expression found in skin cutaneous melanoma (SKCM). (**C**) Pathological stage plot of *LADON* expression in SKCM. (**D**) The overall survival plot of SKCM patients with tumours expressing high or low levels of *LADON*. The hazard ratio (HR) calculation is based on the Cox PH model. (**E**) The RT-PCR with primers (L1F-L1R) amplifying a 130 nt band from the 5′ regions of *LADON* detects the transcript in all cell lines of the panel: melanocytes, non-metastatic melanoma (MNT1), metastatic melanoma (A375, 888 mel, SLM8, FO1), breast cancer (MCF7), and an embryonic kidney. (HEK293). (**F**) The RT-qPCR measurements of the *LADON* expression were performed in keratinocytes (HaCat), melanocytes, non-metastatic (MNT1, DAJU, SK28) or metastatic melanoma cells (A375, FO1, 888 Mel, and SLM8), after 24 h and 96 h of culture. The *LADON* expression was normalized to that of endogenous *RPL13*. For each cell line, the value at 24 h was then set to 1. (**G**) The RT-qPCR measurements (dots) of *LADON* expression after 24 h culture were performed on the same panel as in (**F**) but normalized to the endogenous RPL13 expression. The histograms display mean values ± SD from a minimum of two independent replicates, *p*-values were calculated using the Student’s *t*-test, ** < 0.01.

**Figure 3 ncrna-09-00071-f003:**
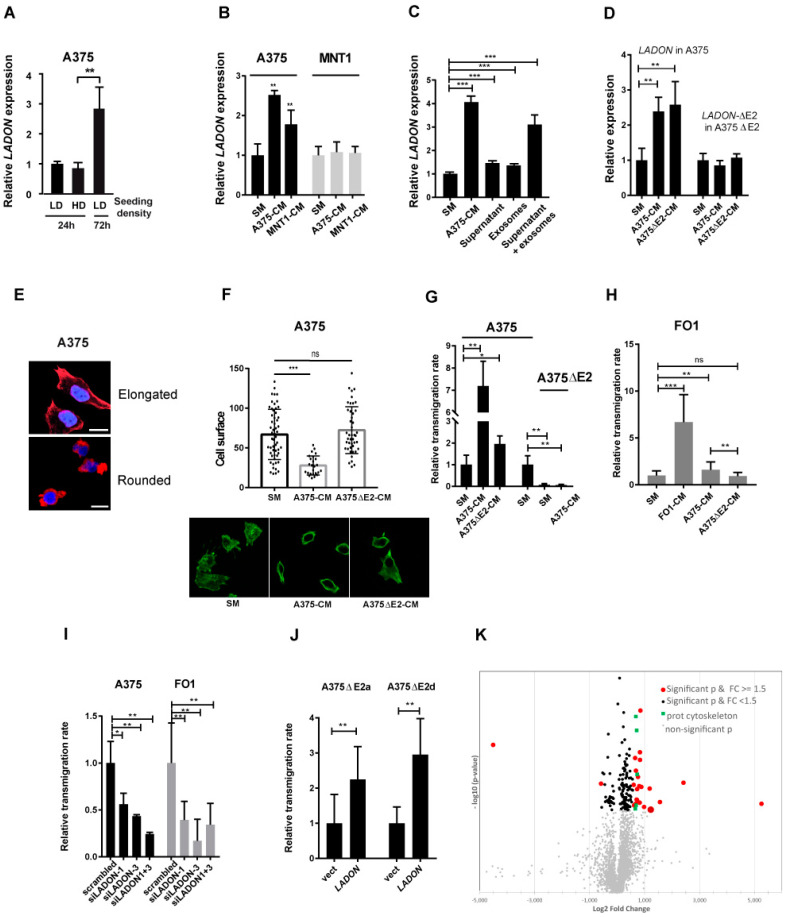
The melanoma cells secrete factors that promote *LADON* expression, and invasion. (**A**) The RT-qPCR analysis of the *LADON* expression in the A375 cells seeded at low (LD) or high density (HD) and cultured for 24 or 72 h. Cell density does not affect *LADON*’s expression level whereas cell culture duration does. The values were normalized to that of the LD condition after 24 h, which was set to 1. (**B**) The RT-qPCR analysis of the *LADON* expression in the A375 and MNT1 cells cultured for 24 h in the standard culture medium (SM), A375-conditioned medium (A375-CM) or MNT1- conditioned medium (MNT1-CM). The values were normalized to that of the SM condition, which was set at 1. (**C**) The RT-qPCR analysis of the *LADON* expression in the A375 cells cultured for 24 h in the SM, A375-CM, supernatant fraction of A375-CM, exosome fraction of A375-CM or a combination of these two fractions. The values were normalized to that of the SM condition, which was set at 1. (**D**) The RT-qPCR analysis of the *LADON* and *LADON-ΔE2* expression in the A375 or A375ΔE2a cells cultured for 24 h in the SM, A375-CM or A375ΔE2-CM. The values were normalized to that of the SM condition, which was set to 1. (**E**) Representative images of the A375 cells with elongated or rounded morphology, visualized after F-actin (red) and Hoechst 33342 (nuclear, blue) staining. Scale bar 25μm. (**F**) Histogram of cell surface measurements of A375 cells cultured for 24 h in SM, A375-CM, or A375ΔE2-CM (*n* > 20) from at least 3 independent experiments. Representative fields are shown under the graph. (**G**,**H**) Relative transmigration rates of A375, FO1, or A375ΔE2a cells cultured in SM, A375-CM, FO1-CM, or A375ΔE2-CM. Values were normalized to that of the SM condition, which was set to 1. (**I**) Relative transmigration rates of A375 or FO1 cells in the presence of the siRNAs scrambled (as control), siLADON-1, siLADON-3, or siLADON-1 and -3. Values were normalized to that obtained with the scrambled siRNA, which was set to 1. (**J**) Relative transmigration rates of A375ΔE2a and A375ΔE2d cells transfected with GFP alone (vector) or together with *LADON*. Values are normalized to those obtained with GFP alone, which were set to 1. Histograms display mean values ± SD from a minimum of three independent replicates. *p*-values were calculated using the Student’s *t*-test, * < 0.05, ** < 0.01, ***<0.001, ns non significant. (**K**) Volcano plot representation of a comparative proteomics analysis between A375 and A375ΔE2 cells. Grey dots: non-significant *p*-value. Black dots, significant *p*-value and fold change <1.5. Red dots: significant *p*-value and fold change >1.5. Green dots, cytoskeleton proteins significant *p*-value and fold change >1.5.

**Figure 4 ncrna-09-00071-f004:**
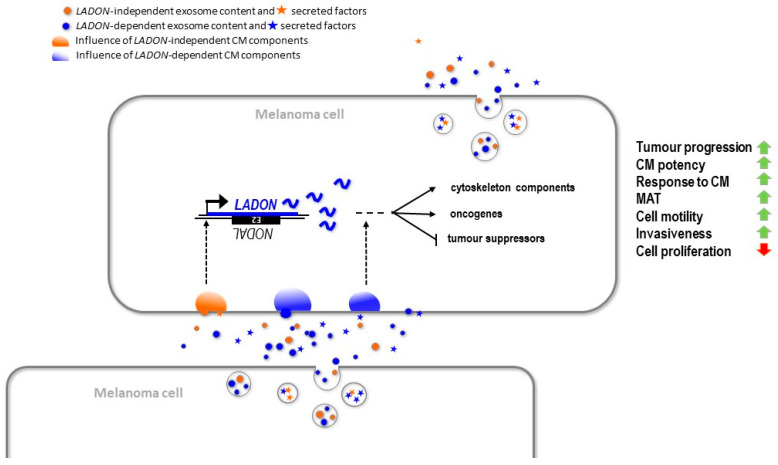
Model for the role of *LADON* in melanoma tumour progression and metastasis. In metastatic melanoma cells, the *LADON* transcript (in blue) promotes the transition from a proliferative cell identity to a less proliferative and more invasive cell identity. In the course of a 4-day culture, an increase of *LADON* expression promotes the expression of known oncogenes and represses that of tumour or metastasis suppressor, which in turn affects the expression of downstream targets, such as cytoskeleton components. The *LADON* expression also affects the potency of the conditioned medium produced by the cells, presumably via its impact on the exosomes (quantity, content) and the secreted factors they release. The *LADON*-dependent exosomes and secreted factors (in blue) promote the progression of exposed melanoma cells toward their more invasive identity. The *LADON* transcript is itself enriched in exosomes.

**Table 1 ncrna-09-00071-t001:** Proteins were significantly enriched using the Gene Ontology approach.

Protein	Ratio Changes A375ΔE2 vs. A375	(*p*) ANOVA	Filamentous Actin	Stress Fiber	Lamellipodium	Accession
Dynactin subunit 4 (DCTN4)	1.664529632	0.01351308		■		Q9UJW0
1phosphatidylinositol 4,5-bisphosphate phosphodiesterase gamma-1 (PLCG1)	1.637021061	0.00271601			■	19174
Fermitin family homolog 2 (FERMT2)	1.618822307	0.0415036	■	■	■	Q96AC1
PDZ and LIM domain protein (PDLIM4)	1.606767329	0.0016411	■	■	■	P50479
Nck-associated protein 1 (NCKAP1)	1.580463985	0.047219	■		■	Q9Y2A7

■ belonging to sub class of cytoskeleton protein.

**Table 2 ncrna-09-00071-t002:** Proteins with a fold change superior to 2.

Protein	Ratio Changes A375ΔE2 vs. A375	(p) ANOVA	Fonction	Accession
Prostatic acid phosphatase (ACPP)	38.17912767	0.03938661	tumor suppressor, regulated by p65	P15309
Transcription factor p65 (RELA)	5.3358974	0.01828842	transcription factor, cell proliferation, apoptosis, and oncogenesis	Q04206
Gamma-aminobutyric acid receptor-associated protein-like2 (GABARAPL2)	2.947432444	0.03702373	Ubiquitin-like modifier involved in intra-Golgi traffic	P60520
Protein FAM107B (FAM107B)	2.274639991	0.02268322	tumor suppressor	Q9H098
Protein NDRG1 (NDRG1)	2.33790023	0.04885314	metastasis suppressor	Q92597
Serine/threonine-protein phosphatase 4 regulatory subunit 2 (PPP4R2)	−22.78074964	0.03702373	overexpression in cancer promotes cell growth and invasion	P60520

**Table 3 ncrna-09-00071-t003:** Reagents or resource.

Chemicals, Peptides, and Recombinant Proteins	Source	Identifier
D-MEM/F-12 (1:1) (1X), liquid plus Glutamax,	LifeTechnologies Carlsbad, CA, USA	31331028
FBS Foetal Bovine Serum, Origin: E.U. Approved (South American)	Life Technologies	10270106
Penicillin/Streptomycin	Life Technologies	15140-148
Trypsin 0.5% EDTA	Thermo Fisher Waltham, MA, USA, États-Unis	25300054
DAPI (4′, 6-diamidino-2-phenylindole)	Invitrogen Molecular Probes (Thermo Fisher)	D1306
ACTIVIN	CELL guidance systems	GFM29
SB431542	Millipore, Guyancourt, France	616461
Dharmafect 1	Dharmacon, Lafayette, CO, USA	T-2001
NODAL	Biotechne, Minneapolis, MN USA	1315-ND-025
Experimental Models: Cell Lines		
A375M	American Type Culture Collection, Manassas, VA, USA	cat# ATCC CRL-1619
Melanocyte	From Nathalie Andrieu	N/A
MNT1, FO1	From Lionel Larue	N/A
SLM8	From Manuelle Viguier	N/A
888 Mel	From Alain Mauviel	N/A
Oligonucleotides		
Primers	See Appendix A	N/A
ON-TARGETplus Non-targeting Control siRNAs	Dharmacon	D-001810-01-05
ON-TARGETplus LADON siRNAs 1,2 and 3	Dharmacon	SO-2770618G
Recombinant DNA/Plasmids		
pcDNA3.1	Thermo Fisher Scientific	V79520
pcDNA3.1 GFP	Thermo Fisher Scientific	
pcDDA3.1 LADON full-length	This paper	N/A
Critical Commercial Assays		
SUPERSCRIPT III VILO (50 réactions)	Thermo Fisher Scientific	11755050
LightCycler FastStart DNA Master SYBR Green I	Roche	12239264001
RiboCellin	EUROBIO, Les Ulis France	RC1000
Software and Algorithms		
Graph Prism	GraphPad Software	Graphpad Prism 10.0.3
Adobe Illustrator	Adobe	Adobe illustrator CS3
Image J	[50]	ImageJ
UCSC	[51]	UCSC
Excel	Microsoft	Microsoft 2019
CFX Maestro	Bio-Rad	CFX Maestro
GPP sgRNA Design	[52]	portals.broadinstitute.org/gppx/crispick/public

## Data Availability

The data set generated for this study can be accessed at Dryad DOI https://datadryad.org/stash/share/y--gzkcuMGyMH3nf1enFttCDwl9eIxFt0Ob3daUCzTY (accessed on 9 October 2023).

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
