# Peer review of "LADON, a Natural Antisense Transcript of NODAL, Promotes Tumour Progression and Metastasis in Melanoma"

_ncrna, 2023, doi:10.3390/ncrna9060071_

Round 1

Reviewer 1 Report

Comments and Suggestions for Authors

The manuscript by Dutriaux et al. is of moderate to high significance to ncRNA readership. The model and techniques used are appropriate, and the focus on natural antisense transcripts is relevant to the field. Also, the choice of models for melanoma is fitting. 

Several issues need to be addressed:

- the overall quality of delivery and flow should be improved; some paragraphs seem wordy, while some are too concise

- figure S1B is of poor quality, it is unclear whether the exon1/2 product was detected

minor issues:

- the image in figure S1C lacks the lane labeling

- the sentence in line 68 is unfinished

Comments on the Quality of English Language

no major issues

Reviewer 2 Report

Comments and Suggestions for Authors

In the manuscript by Dutriaux et al., the authors examined the involvement of LADON, antisense transcripts in facilitating the process of tumour progression and apoptosis in melanoma.

The data presented in the manuscript is interesting. The manuscript is well written, and the experiments are well designed. Some further experiments need to be done in order to explore the gene regulation associated with the event.

I have a few major/minor suggestions for polishing the manuscript.

Major Comments:

1.     The authors state that LADON antisense transcripts are responsible for the metastatic behaviour of the melanoma cell lines. The authors should do a soft agar assay to test the anchorage independent growth with A375 and A375∆E2 cells lacking the truncated LADON transcript.  The experiment will re-validate the hypothesis that is truly responsible for metastatic behaviour.

2.     In Fig 3A, the authors should also check the expression of LADON transcripts in the HD 72hr A375 cells. This data will also act as a control.

3.     The authors states that melanoma cells secrete factors that aid in LADON expression.  The authors should perform comparative proteomic analysis in A375 and A375∆E2 cells.  This experiment will further uncover additional protein factors that aid in the expression of LADON transcripts mediating tumour progression.

4.     The authors should include representative microscopic images in Fig 3F,3G and SD-F.

5.     The authors should include the roundness index of FO1 cells in Supplementary Fig3.

6.     The authors can include a model graphical figure for the broader audience.

Minor Comments

1.     The abstract should be rewritten.  E.g., Lines 11 and 18 can be combined together to make the abstract much more interesting. The overall manuscript is well written, but rewriting the abstract will improve it further.

2.     Page 2 Line 68 before Ref. 13 ‘and further suggest that’ - the sentence is incomplete.

3.     In Fig S1B, there is a faint band noted in MNT1 that corresponds to the NODAL exon 2-3 junctional region.  Is that a true band or a non-specific band?

4.     In Figure S1C, the authors should mention from the figure which clones of A375∆E2 they chose for their experiment.

5.     The authors should add some extra lines explaining the results of Fig 1C and S1F.

6.     In Fig 3C, is the relative expression of LADON transcripts in supernatant and exosomes really significant. Comment on this.

7.     In Fig S3B, relative LADON expression is reduced in the CM fractions of DAJU and SK28 cells. The authors should address this significant change in reduction of LADON expression.

8.     In the Materials and Methods Section, the authors should state which procedure they used to calculate the relative gene expression through real time PCR such as 2∆∆ct or any other method.

Round 2

Reviewer 2 Report

Comments and Suggestions for Authors

In the manuscript by Dutriaux et al., the authors examined the involvement of LADON, antisense transcripts in facilitating the process of tumour progression and apoptosis in melanoma.

The authors have addressed all the previous comments. Thus, the manuscript can be accepted in its present form.